# DATS: Difficulty-Aware Task Sampler for Meta-Learning Physics-Informed Neural Networks

**Maryam Toloubidokhti**[*1]**, Yubo Ye** [*2]**, Ryan Missel**[1]**, Xiajun Jiang**[1]**, Nilesh Kumar**[1]**,
Ruby Shrestha**[1] **& Linwei Wang**[1]
[1]Rochester Institute of Technology, Rochester, NY, USA
[2]Zhejiang University, Hangzhou, China
{mt6129}@rit.edu, {22230131}@zju.edu.cn

## Abstract

Advancements in deep learning have led to the development of physics-informed neural networks (PINNs) for solving partial differential equations (PDEs) without being supervised by PDE solutions. While vanilla PINNs require training one network per PDE configuration, recent works have showed the potential to meta-learn PINNs across a range of PDE configurations. It is however known that PINN training is associated with different levels of difficulty, depending on the underlying PDE configurations or the number of residual sampling points available. Existing meta-learning approaches, however, treat all PINN tasks equally. We address this gap by introducing a novel difficulty-aware task sampler (DATS) for meta-learning of PINNs. We derive an optimal analytical solution to optimize the probability for sampling individual PINN tasks in order to minimize their validation loss across tasks. We further present two alternative strategies to utilize this sampling probability to either adaptively weigh PINN tasks, or dynamically allocate optimal residual points across tasks. We evaluated DATS against uniform and self-paced task-sampling baselines on two representative meta-PINN models, across five benchmark PDEs as well as three different residual point sampling strategies. The results demonstrated that DATS was able to improve the accuracy of meta-learned PINN solutions when reducing performance disparity across PDE configurations, at only a fraction of residual sampling budgets required by its baselines [1].

## 1 Introduction

Partial differential equations (PDEs) underpin a broad range of scientific simulations such as fluid dynamics, heat transfer, and electromagnetics (32). While traditional methods for solving PDEs can be numerically and computationally challenging over high-dimensional and complex domains (2), these challenges are being addressed by recent advancements in deep learning (13). Particularly of note are the physics-informed neural networks (PINNs) that integrate the governing PDEs as a loss function to train a neural network, approximating PDE solutions without discretizing the underlying solution domain while improving efficiency (30). Because the PINN is supervised by the governing PDE equation, its training is *unsupervised* without requiring PDE solutions available (8; 30).

However, incorporating governing PDE equations in the loss function also leads to one of the fundamental limitations in PINNs: it has to be re-trained each time when the configuration of the underlying PDE changes (30; 8). There have been increasing interests in addressing this limitation (28; 29; 1; 16; 36). Most recent successes have been built on the meta-learning concept, aiming to learn a meta-model across a range of PDE configurations such that it can either directly generate the PINN for a given PDE configuration (1), or can be rapidly fine-tuned to a given PDE configuration (16; 36; 22). Treating PINN-training for each PDE configuration as one task, these meta-models are trained across a set of tasks uniformly sampled from a range of PDE configurations.

When training the PINN, however, it is known that the difficulty for solving the underlying PDEs varies. Examples of determinants for such task difficulty include: 1) the value of the PDE parameters

---

[*]Both authors contributed equally to this work
[1]Source code available at https://github.com/maryamTolou/DATS_ICLR2024.

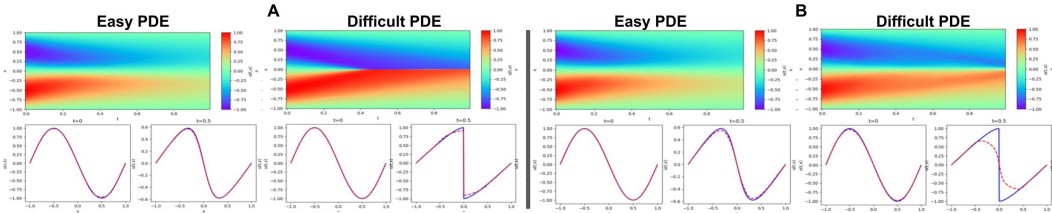

Figure 1: PINN solutions for the Burger equation with low *versus* high Reynolds numbers, obtained at a higher budget (A) *versus* a lower budget (B) of residual points for training.

or initial conditions being considered (19), and 2) the available budget of sampling points on which PDE residual loss can be calculated (commonly referred to "residual points") (34). For instance, Burger equations with high Reynolds numbers are associated with PDE solutions with sharp transitions as illustrated in Fig. 1A, presenting more significant challenges for PINN to solve (19). As the sampling budget of residual points decreases, the difficulty of PINN training increases, although much more significantly for the higher Reynolds numbers (Fig. 1B). The existing meta-learning approaches to PINNs – which sample uniformly across all PDE configurations each with the same budget of residual points (28; 29; 1; 16; 36) – neglect such differences in task difficulty. This can lead to degraded average performance as well as performance disparity among various PDE configurations.

In this paper, we introduce a novel difficulty-aware task sampler (DATS) for meta-learning of PINNs, with a goal to prioritize different PINN tasks depending on the task difficulty associated with the underlying PDE configurations. We are motivated by the recently presented concepts in general meta-learning that optimize task sampling probabilities during *meta-training* to minimize the average performance of all tasks during *meta-validation* (35). While recent works approach this through neural optimization procedures realized via reinforncement learning (23; 35), we theoretically establish analytical solutions for the optimal task sampling probability across the range of PDE configurations considered. In the practical setting where a discrete set of PDE configurations is considered, we then present a novel strategy to convert this sampling probability to dynamic allocations of residual points among different PDE configurations (DATS-rp), and contrast it with a simpler strategy that directly weighs the contribution of different tasks to the meta-training loss (DATS-w).

We note that there is not yet any existing adaptive task sampling strategy for meta-learning PINNs. While related concepts exist in general meta-learning, they are not trivial to directly extend to the PINN due to its unique *unsupervised* learning nature. In parallel, adaptive sampling strategies are being studied in PINN literature but at the level of sampling "residual points" for training a single PINN (34; 27; 9; 25) – one representative approach is based on self-paced learning that automatically schedules the sampling of training residual points based on the difficulty level of obtaining PDE solutions at those points (11). As this is relevant to the concept of "difficulty-aware sampling" in DATS, we adopt the methodology of self-paced learning (20) described in (11) but extend it (from the original level of sampling residual points) to the level of task sampling in meta-learning PINN: this gives us two primary baselines for evaluating DATS: uniform task sampling routinely used in meta-learning PINNs, and self-paced task sampling as a stronger adaptive task sampling baseline.

To demonstrate that the benefits of DATS task-level sampling is orthogonal to residual-point sampling strategies used for learning each PINN, we evaluate DATS and its two task-sampling baselines in combination with three representative residual point sampling strategies (34). To further demonstrate that DATS is agnostic to the underlying choices of meta-learning approaches, we conduct experiments on two representative existing meta-PINNs (1; 16). We conduct these experiments in four benchmark PDEs commonly used in PINN literature and, to demonstrate the importance of considering task difficulty in meta-PINNs, we consider ranges of PDE parameters and residual-point budgets beyond those commonly used in literature. We further demonstrate that DATS can be used to meta-learn across more general PDE configurations such as varying initial conditions.

Throughout all benchmark PDEs, meta-PINN frameworks, and residual-point sampling strategies considered, DATS demonstrated its ability to significantly improve the average performance of all PINN tasks while reducing their performance disparity, all achieved at a substantially smaller budgets of residual points in comparison to uniform or self-paced task-sampling in meta-learning PINNs.

## 2 RELATED WORKS

**Meta Learning of PINNs:** A PINN is typically trained to solve a PDE with a given configuration. By considering different PDE configurations as different tasks, there is a recent interest in leveraging meta learning techniques to approximate the solution manifold over the PDE configuration space (28). Recent meta-PINN approaches can be categorized based on whether it leverages feedforward- or agnostic meta-learning (MAML) based meta-learning frameworks. In feedforward-based meta-PINNs, a meta-model is learned to map the PDE configurations to PINN weight parameters (1; 6). This mapping is learned as a hypernetwork (12) in HyperPINN (1), while learned with supervision by the data pairs of PDE configurations and PINN weights in (28). Meta-MgNet (7) introduces a hypernetwork within the Multi-grid Network (MgNet). In MAML-based meta-PINNs, an effective initialization for the PINN weight parameters is learned such that, with a few gradient updates, the network can be fine-tuned to a new given PDE configuration (22; 16). This is achieved in (22) by a Reptile-based strategy to directly learn the initialization of the main PINN. In MAD-PINN (16), a latent embedding is introduced as an additional input to the main PINN and is fine-tuned to represent an *implicit* encoding of individual PDE configurations. In (36), MAML is applied to train and fine-tune a meta-PINN for 1D parametric arc models. Until now, all these meta-PINN works treat all individual PDE tasks/configurations equally, neglecting the potential differences in task difficulty associated with the PDE configurations and their impact on the meta-model. DATS is the first task-sampling strategy for meta-PINNs and is agnostic to the type of meta-learning approaches used: to demonstrate this, we evaluate DATS on two meta-PINN models representing each of the feedforward- and MAML-based approaches: HyperPINN (1) and MAD-PINN (16).

A recently introduced GPT-PINN (5) solves a similar problem by extending the idea of reduced order model: it learns to use pre-trained PINNs for a finite set of PDE configurations as activation functions in the PINN of a new PDE. To determine when to add a neuron for a given PDE configuration inherently considers its difficulty. We thus included it as a baseline in a subset of experiments.

**Adaptive Residual Sampling in PINN:** For training an individual PINN, the most commonly used strategies for selecting residual points is uniform random sampling (34) from a uniform distribution or equi-spaced grid. There however has been an increasing interest in adaptive sampling techniques for residual points (34; 9). Examples include the residual-based adaptive refinement (RAR)(25) to sample more residual points in areas with significant PDE losses, and residual-based adaptive distribution (RAD) that resamples residual points from a probability mass function defined over the PDE loss (34). Recently, self-paced learning is also introduced to enable adaptive residual point sampling by automatically scheduling their inclusion in training based on their PDE losses (11).

Despite the shared motivation in difficulty-aware sampling, adaptive residual sampling and DATS are fundamentally different: the latter addresses task-level sampling when meta-learning PINNs, while the former addresses sampling at the level of residual points when learning a single PINN. To understand how the benefits of task sampling may be influenced by the lower-level residual points sampling, we will evaluate DATS and its task-sampling baselines in combination with three residual point sampling strategies: random, random with resampling (random-R) (25), and RAD that was reported as the best performing residual point sampling technique in a recent survey (34).

**Adaptive Task Sampling in General Meta Learning:** In general, it is increasingly recognized that different tasks may have different learning difficulties with different impact on the meta-model (4). To address this, a probabilistic active learning method is designed in (18) to rank the latent task embeddings using a utility function. In (21), a task difficulty measure is defined over pairs of classes to define an evolving task selection probability. In (35; 21), tasks are prioritized based on their contribution to the generalization of the meta-model at meta-validation, realized via reinforncement learning. None of these task measures or optimization strategies is trivial to extend to meta-PINNs.

DATS shares the motivation of (21; 35) but approaches the bi-level optimization with a theoretically backed analytical solution, establishing the first adaptive task sampler for meta-learning PINNs. While there is not an existing baseline nor a trivial extension that can be achieved from general meta-learning, the self-paced adaptive sampling of PINN residual points mentioned earlier (11) shares some high-level similarity to DATS. We thus extend (11) – originally designed for residual point sampling – to task sampling when meta-learning PINNs. This establishes an adaptive task-sampling baseline for DATS, with a key difference that DATS optimizes task sampling to minimize the meta-model's validation loss, whereas self-paced learning does so to minimizes the meta-model's training loss.

**Alternative Deep Learning Approaches to Solving PDEs:** There are alternative deep learning techniques for solving parametric partial differential equations, such as operator learning (*e.g.*, DeepONet (24)) that, unlike unsupervised PINN trianing, often relies on supervision from explicit PDE solutions. Emerging works have also attempted to combine these two approaches resulting in physics-informed DeepONets (PIDeepONets) (33). As DATS is designed to address challenges associated with meta-learning unsupervised PINNs, comparison with other *supervised* PDE-solving solutions is outside the scope of this work.

## 3  PRELIMINARIES

**Learning PDEs with PINNs:** PINNs approximate the solution $u(x)$ to a PDE, by training a neural network $\hat{u}_\phi(x)$ parameterized by $\phi$ to satisfy the given PDE and boundary/initial conditions. Assuming a PDE in the general form of $F(u, \nabla u, \nabla^2 u, \ldots, x, \lambda)$, the PINN loss function $\mathcal{L}_{\text{PINN}}(x, \hat{u}_\phi, \lambda) = \mathcal{L}_{\text{BI}} + \mathcal{L}_{\text{Residual}}$ includes two terms:

$$\mathcal{L}_{\text{Residual}}(x, \hat{u}_\phi, \lambda) = |F(\hat{u}_\phi, \nabla \hat{u}_\phi, \nabla^2 \hat{u}_\phi, \ldots, x, \lambda)|^2 \quad \text{where} \quad x \in P_r \tag{1}$$

$$\mathcal{L}_{\text{BI}}(x, \hat{u}_\phi, \lambda) = |u(x) - \hat{u}_\phi(x)|^2 \text{ or } \left|\frac{\partial \hat{u}_\phi}{\partial n} - g(u)\right|^2, \text{ where } x = 0 \text{ or } x \in \partial B \tag{2}$$

where $L_{\text{Residual}}$, tied to a specific value of the PDE parameter $\lambda$, computes the PDE residual on a set of residual points $x \in P_r$ sampled from the solution domain without knowing the corresponding PDE solutions. $\mathcal{L}_{\text{BI}}$ defines the boundary/initial conditions . As the PDE parameter or boundary/initial conditions change, the PINN has to be re-trained from scratch, a main challenge for training PINNs for parametric PDEs over a range of configuration values or boundary/initial conditions.

**Meta-Learning PINNs:** Meta-learning has been increasingly applied to PINNs to learn a meta model, with meta-parameters $\theta$, that can quickly adapt a PINN $\hat{u}_\phi(x; \theta)$ to any given PDE configuration $\lambda \sim p(\lambda)$. We consider two representative meta-PINNs: HyperPINN (1) and MAD-PINN (16).

HyperPINN (1) formulates the meta-model as a hyper-network that maps PDE configuration $\lambda$ to the weight parameters $\phi$ of a main PINN as: $\phi = H_\theta(\lambda)$, such that $\hat{u}_\phi(x; \theta) = \hat{u}_{H_\theta(\lambda)}(x)$. The hyper-network is optimized across a range of PDE configurations with:

$$\theta^* = \arg\min_\theta \mathbb{E}_{\lambda \sim p(\lambda)} \left[\mathcal{L}_{\text{PINN}}(x, \hat{u}_{H_\theta(\lambda)}, \lambda)\right] \quad : \quad x \in P_r \tag{3}$$

MAD-PINN (16), in contrast, provides an additional input $z_\lambda$ to a regular PINN as $\hat{u}_\phi(x; \theta) = \hat{u}_\phi(x, z_\lambda)$. $z_\lambda$, termed as *implicit code*, is a vector that is optimized individually for each PDE configuration $\lambda$ along with the weight configuration $\phi$ of the PINN, over a range of $\lambda$ via:

$$\theta^* = (\{z_\lambda^*\}, \phi^*) = \arg\min_{\phi, z_\lambda} \mathbb{E}_{\lambda \sim p(\lambda)}[\mathcal{L}_{\text{PINN}}(x, \hat{u}_\phi(x, z_\lambda), \lambda) + \frac{1}{\sigma^2}\|z_\lambda\|^2] \quad : \quad x \in P_r \tag{4}$$

where $\frac{1}{\sigma^2}\|z_{\lambda_i}\|^2$ enforces training stability and $\sigma$ is a hyper-parameter. When a new PDE configuration arises, MAD-PINN achieves fast transfer by fine-tuning $z_\lambda$ (or along with $\phi$) to the new task.

Therefore, HyperPINN resembles a feed-forward meta-learning approach (15), whereas MAD-PINN resembles a MAML approach (10). HyperPINN does not require test-time fine-tuning, although it considers meta-training only across PDE parameters whereas MAD-PINN can accommodate broader definitions of PDE configurations. They thus provide two diverse settings to test DATS.

## 4  METHODOLOGY

In existing meta-PINN works, optimization of equation 3 or equation 4 is done over a range of $\lambda$'s sampled from a *uniform* distribution of $p(\lambda)$, with the same number of residual points used in each task. This treats all PDE configurations equally and does not take into account the difficulty levels of solving different PDEs. Instead, we propose DATS to optimize the sampling probability $p(\lambda)$ for meta-training, such that the resulting model will minimize the average validation loss as follows:

$$p^*(\lambda) = \arg\min_{p(\lambda)} \mathbb{E}_{\lambda \sim \mathcal{U}(\lambda)}\{\mathcal{L}_{\text{PINN}}(x, \hat{u}_\phi(x; \theta^*), \lambda)\} \quad : \quad x \in P_{val},$$

$$\hat{u}_\phi(x; \theta^*) = \begin{cases} \hat{u}_{H_{\theta^*}}(x) & \theta^* = \text{solution of 3} \\ \hat{u_{\phi^*}}(x, z_\lambda^*) & \theta^* = (\{z_\lambda^*\}, \phi^*) = \text{solution of 4} \end{cases} \tag{5}$$

Here, $\mathcal{L}_{\text{PINN}}(x, \hat{u}_\phi(x; \theta^*), \lambda)$ denotes the loss of the PINN $\hat{u}_\phi(x; \theta^*)$ on meta-validation residual points $x \in P_{val}$, calculated over uniformly sampled tasks $\lambda \sim \mathcal{U}(\lambda)$. The individual PINNs $\hat{u}_\phi(x; \theta^*)$ are optimized from the meta-learning objective (equation 3 or equation 4), over the meta-training task sampling probability $p(\lambda)$. This introduces a nested bi-level optimization where the optimization of $p(\lambda)$ in equation 5 is defined over the optimization of equation 3 or equation 4 over the given $p(\lambda)$. Because of this nested optimization, existing works (23; 35) have resorted to reinforcement learning to solve equation 5. To reduce this complexity, we derive analytical solutions to equation 5 for DATS.

## 4.1 ANALYTICAL SOLUTIONS

For readability, below we denote the validation loss in equation 5 as $l_{val,\lambda}$, and the training loss as $l_{tr,\lambda}$. In an iterative optimization scheme to solve equation 5, instead of actually optimizing $\hat{u}_\phi(x; \theta^{t+1})$ over $p(\lambda)^t$ via equation 3 or equation 4 at each iteration $t$, we note that $\theta^{t+1}$ can be approximated by a single-step of gradient descent as $\theta^{t+1} = \theta^t - \eta \int_\lambda p(\lambda) \nabla_\theta l_{tr,\lambda}(\theta^t) \, d\lambda$. With this, the validation loss at iteration $t + 1$, $l_{val,\lambda}(\theta^{t+1})$, can be expressed as $l_{val,\lambda}(\theta^t - \eta \int_\lambda p(\lambda) \nabla_\theta l_{tr,\lambda}(\theta^t) \, d\lambda)$ which, with first-order Taylor expansion, can be expressed as:

$$l_{val,\lambda}(\theta^t) - \eta \int_\lambda p(\lambda) \nabla_\theta l_{tr,\lambda}(\theta^t) d\lambda \cdot \nabla_\theta l_{val,\lambda}(\theta^t) \tag{6}$$

Substituting equation 6 into equation 5, and letting $\nabla_\theta l_{tr,\lambda} = g_{tr,\lambda}$ and $\nabla_\theta l_{val,\lambda} = g_{val,\lambda}$, we have:

$$
\begin{aligned}
p^{t+1}(\lambda) &= \arg\min_{p(\lambda)} \mathbb{E}_{\lambda \sim \mathcal{U}(\lambda)} \{ l_{val,\lambda}(\theta^t) - \eta \int_\lambda p(\lambda) g_{tr,\lambda}(\theta^t) d\lambda \cdot g_{val,\lambda}(\theta^t) \} \\
&= \arg\max_{p(\lambda)} \mathbb{E}_{\lambda \sim \mathcal{U}(\lambda)} \{ ( \underline{\int_\lambda p(\lambda) g_{tr,\lambda}(\theta^t) d\lambda} ) \cdot g_{val,\lambda}(\theta^t) \}
\end{aligned}
\tag{7}
$$

where the underlined integral represents expectation of the training gradient over $p(\lambda)$, and the following term represents the validation gradient on a particular value of $\lambda$. This use of first-order approximation in equation 6 and assumption of the single-step gradient descent for the optimization nested within is inspired by the first-order meta-learning algorithms described in Reptile (26). It allows equation 7 to be optimized analytically when iterating with the optimization of the meta-PINNs: at iteration $t$, task-sampling probability $p^t(\lambda)$ is first optimized by equation 7 using the current training and validation loss of all task $\lambda$'s; the meta-model $\theta^{t+1}$ is then updated given $p^t(\lambda)$.

**Regularization Strategies:** To stablize task probability assignments, we further regularize $p(\lambda)$ to a prior distribution $r(\lambda)$ with the Kullback–Leibler (KL) divergence and its strength modulated by a hyperparameter $\beta$. We consider two options for $r(\lambda)$: 1) $r(\lambda) = \mathcal{U}(r)$ as a uniform prior (*uniform-KL*), or 2) $r(\lambda) = p(\lambda^t)$ optimized at the previous iteration $t$ (*consecutive-KL*). We will ablate these two regularization strategies in experimental evaluations.

## 4.2 DISCRETE APPROXIMATIONS

Considering a typical setting of meta-learning PINNs with a discrete set of PDE configurations $\{\lambda_1, \lambda_2, ..., \lambda_n\}$, we further obtain a discrete approximation of the objective function in equation 7:

$$\frac{1}{n} \sum_{j=1}^n ((\sum_{i=1}^n p(\lambda_i) g_{tr,\lambda_i}(\theta^t)) \cdot g_{val,\lambda_j}(\theta^t)) = \frac{1}{n} \sum_{i=1}^n p(\lambda_i) w_i, w_i = \langle g_{tr,\lambda_i}(\theta^t), \sum_{j=1}^n g_{val,\lambda_j}(\theta^t) \rangle \tag{8}$$

where $\langle, \rangle$ denotes the dot product. Intuitivtely, $w_i$ measures the gradient similarity between the training loss of task $\lambda_i$ and the validation loss across all tasks. This results in a higher $w_i$ to a PDE configuration $\lambda_i$ that is most beneficial for reducing the valudation loss across all tasks. Based on Karush–Kuhn–Tucker (KKT) first-order necessary condition for an optimal solution, we can then derive an analytical solution to our optimization as:

$$p^{t+1}(\lambda_i) = r_i * exp(\frac{1}{\beta} w_i) / \sum_i r_i * exp(\frac{1}{\beta} w_i) \tag{9}$$

where $r_i = \frac{1}{n}$ for uniform-KL regularization, and $r_i = p^t(\lambda_i)$ for consecutive-KL regularization (derivations are included in Appendix A). Considering a total budget of $b_T$ residual points and a task-specific budget of $b_{\lambda_i}$, we now introduce two strategies to utilize $p^{t+1}(\lambda_i)$ to realize the meta-PINN training objective in equation 3 or equation 4 over the discrete representations of $\lambda$.

**DATS-w – Adaptive Weighting of Meta-Training Losses:** As $p^{t+1}(\lambda_i)$ determines the sampling probability for PDE configuration $\lambda_i$, the most intuitive solution based on importance weighting would be to interpret $p^{t+1}(\lambda_i)$ as adaptive weighting to control the contribution of each PDE task $i$ to the training loss. The meta-training objectives in equation 3 or equation 4 will then become:

$$\theta^{t+1} = \arg\min_\theta \frac{1}{b_T} \sum_{i=1}^{n} p^{t+1}(\lambda_i) \sum_{k=1}^{b_T/n} l_{tr,\lambda_i}(\theta, x_{\lambda_i,k}), \quad x_{\lambda_i,k} \in P_{\lambda_i,r}, |P_{\lambda_i,r}| = b_T/n \quad (10)$$

where $P_{\lambda_i,r}$ denotes the training residual points for PDE configuration $\lambda_i$, and $|P_{\lambda_i,r}|$ denotes it size: in another word, the budget of residual points is assigned equally across all tasks as in existing works.

**DATS-rp – Adaptive Allocation of Residual Points:** Considering the importance of residual points for PINN training, we consider a novel alternative that utilizes $p^{t+1}(\lambda_i)$ to control the allocation of resources for each PDE task. Instead of assigning residual points uniformly across all tasks, we will adaptively change the budget of residual point sampling for different tasks via $|P_{\lambda_i,r}| = p^{t+1}(\lambda_i) * b_T$, such that more residual points will be assigned to tasks receiving higher $p^{t+1}(\lambda_i)$. This results in:

$$\theta^{t+1} = \arg\min_\theta \frac{1}{b_T} \sum_{i=1}^{n} \sum_{k=1}^{p^{t+1}(\lambda_i)*b_T} l_{tr,\lambda_i}(\theta, x_{\lambda_i,k}), \quad x_{\lambda_i,k} \in P_{\lambda_i,r}, |P_{\lambda_i,r}| = p^{t+1}(\lambda_i) * b_T \quad (11)$$

In experimental evaluations, we will carry out ablation studies to consider the efficacy of the adaptive weighting *vs.* resource allocation strategies above, along with the two KL-regularization strategies.

## 5 EXPERIMENTS

We evaluated DATS in both HyperPINN (1) and MAD-PINN (16), in comparison with two baselines of task sampling strategies: 1) uniform task sampling as used in all existing works, and 2) self-paced task sampling. Furthermore, on the Burger equation, we considered the additional baseline of GPT-PINN (5). We focused on three categories of evaluations. First, on the Burgers' equation (3) – one of the most commonly used benchmarks in PINNs – we conducted ablation studies to understand the adaptive weighting *vs.* residual-point allocation strategies, the uniform *vs.* consecutive-KL regularization strategies, and the effect of the hyperparameter $\beta$. Second, we carried out an extensive set of comparison studies among DATS, uniform, and self-paced task sampling in the presence of three residual point sampling strategies: random (30; 34), random-R (25), and RAD as described earlier (34). Finally, on five benchmark PDE equations: Burgers' (3), convection (19; 14), reaction-diffusion (19; 14), 2D Helmholtz (31) equations, and 3D Navier-Stokes equation (17) , we performed comprehensive evaluations on DATS *vs.* its baselines across different residual-point sampling budgets. More specifically, we set up the upper and lower bound of the meta-PINN PDE solution errors for the baselines using, respectively, a small and large residual point budget. We then evaluated DATS with a series of budgets in between, testing the hypotheses that 1) DATS will achieve a performance improvements compared to its two baselines at a given budget, and this improvement will be more significant at smaller budgets; and 2) DATS will need a much smaller budget to achieve the same performance of its two baselines. We measured the performance of the meta-PINNs with: 1) *average* L2 error of the PDE solutions, and 2) *performance disparity* among the PDE configurations considered, defined as the performance difference between worst-performing to best-performing PINNs. In addition to computing these metrics for PDE configurations included in the meta-training, which is the most common way PINNs are evaluated, we further examined these metrics when HyperPINN was generalized to new PDE configurations not included in the meta-training. We left MAD-PINN out of the latter test as it relies on fine-tuning to generalize which we expect to confound and also reduce the benefits of DATS. More implementation details are included in Appendix B.

### 5.1 DATS ABLATION STUDIES

We ablated the major components of DATS using the 1D Burgers' equation. The special characteristic of this PDE, *i.e.,* the existence of the shock wave, makes it challenging for PINNs to learn, especially for Burgers' equation with a large Reynolds number (R) such as $\lambda = 1/R \approx 1e-3$. In this experiment, we considered 12 values of $\lambda \in (1e-3, 0.1)$ in meta-training and a total residual budget of $b_T = 1000 \times 12$. The base model used was HyperPINN with the original architecture introduced in

|  |  | L2 Error |  | Disparity |  |
| --- | --- | --- | --- | --- | --- |
| KL | $\beta$ | DATS-rp | DATS-w | DATS-rp | DATS-w |
| **Uniform** | **10** | **0.064 ± 0.009** | 0.151±0.056 | **0.075 ± 0.026** | 0.188±0.134 |
| Uniform | 1 | 0.205 ± 0.075 | 0.147±0.051 | 0.173 ± 0.034 | 0.196±0.041 |
| Uniform | 0.1 | 0.168 ± 0.052 | 0.153±0.087 | 0.188 ± 0.095 | 0.207±-0.128 |
| Consecutive | 10 | 0.233 ± 0.057 | 0.210±0.022 | 0.205 ± 0.029 | 0.116±0.035 |
| Consecutive | 1 | 0.129 ± 0.049 | 0.191±0.095 | 0.179 ± 0.075 | 0.207±0.093 |
| Consecutive | 0.1 | 0.169 ± 0.063 | 0.183±0.103 | 0.179 ± 0.075 | 0.177±0.091 |
| Unifrom Sampling | | 0.248 ± 0.027 | | 0.212 ± 0.043 | |

Table 1: DATS ablation of adaptive weighting *versus* adaptive residual point allocation, and uniform-KL *versus* consecutive-KL regularizations under difference values of the hyperparameter $\beta$.

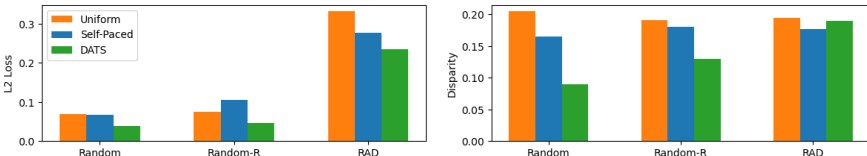

Figure 2: L2 errors (left) and disparity (right) of HyperPINN when using DATS, uniform, and self-paced task sampling in combination with random, random-R, and RAD residual point sampling.

(1). We investigated the effect of 1) uniform-KL *versus* consecutive-KL regularization, and 2) adaptive weighting *versus* adaptive residual-point allocation strategies, with three choices of $\beta \in \{0.1, 1, 10\}$. The results are summarized in Table 1. All variations of DATS outperformed HyperPINN with uniform task sampling. DATS-rp with a uniform KL regularization, using a hyperconfiguration $\beta = 10$, achieved the best performance throughout all experiments. Thus the experiments reported in the rest of the paper considered DATS-rp with a uniform KL regularization, where $\beta$ was tuned using grid search for $\beta \in \{0.1, 1, 10\}$ for each PDE equation separately.

## 5.2 THE EFFECT OF RESIDUAL POINT SAMPLING ON TASK SAMPLING STRATEGIES

We tested how the comparison of DATS and baselines may be affected by the residual point sampling strategies. We continued to use HyperPINN on the Burger's equation for 6 values of $\lambda \in (1e-3, 0.1)$ and a total residual budget of $b_T = 3000 \times 6$. We considered three residual point sampling techniques: 1) random, 2) random with resampling (random-R) (25), and 3) RAD (34). As shown in Figure .2, DATS consistently outperformed the uniform and self-paced baselines, regardless of the residual point sampling strategies (except in the case of RAD, where all three methods obtained comparable disparity metrics). These showed that the effect of adaptive sampling at the task level is more significant than and orthogonal to the effect of adaptive sampling at the level of residual points. Interestingly, contrary to the recent findings on the benefit of adaptive residual point sampling (34), in our experiments random residual sampling was more effective. This suggested that effective residual sampling needed for meta-learning PINNs may be different from that needed for learning a single PINN, revealing a gap of knowledge for further research for the PINN community.

## 5.3 DATS, UNIFORM, SELF-PACED AND GPT-PINN UNDER VARYING BUDGETS

Finally, we investigated the performance of DATS *versus* its uniform and self-paced task sampling baselines under varying budgets of residual points, for both MAD-PINN (16) and HyperPINN (1) across five benchmark PDE equations. For Burger equation, we also considered the additional baseline of GPT-PINN. For each equation, we considered a wide range of their PDE parameters informed by literature. To further demonstrate that DATS is not limited to meta-learning across PDE parameters only, we also considered meta-learning across different initial conditions $u(x, 0) = \alpha_1 sin(\alpha_2 x)$ for the convection equation considering MAD-PINN. HyperPINN is not included in this sub-study as it was only designed to handle varying PDE parameters in its original work. Table 2 summarizes the range and number of PDE configurations considered in each equation.

**Results on Solving PDEs:** Figure 3 summarizes the results for varying PDE parameters across the first four equations and two meta-PINN models. The $x$-axis denotes the maximum residual point

| | PDE | Configuration | #Training | #Test |
|---|---|---|---|---|
| **Burger** | $u_t + uu_x - \lambda u_{xx} = 0$ | $\lambda \in (0.001, 0.1)$ | 14 | 6 |
| **Convection** | $u_t + \beta uu_x = 0 :$ | $\beta \in (0, 10)$ | 5 | 3 |
| | $u(x,0) = a_1 sin(a_2 x)$ | $(a_1, a_2) \in (1, 3)$ | 9 | 4 |
| **Reaction Diffusion** | $u_t - \nu u_{xx} - au(1-u) = 0$ | $\nu, a \in (1, 5)$ | 9 | 4 |
| **Helmholtz (2D)** | $u_{x,y} + k^2 u_{x,y} = q(a,b,x,y)$ | $a_1, a_2 \in (0.5, 1.5)$ | 9 | 4 |

Table 2: The range and number of PDE configurations considered in each PDE benchmark.

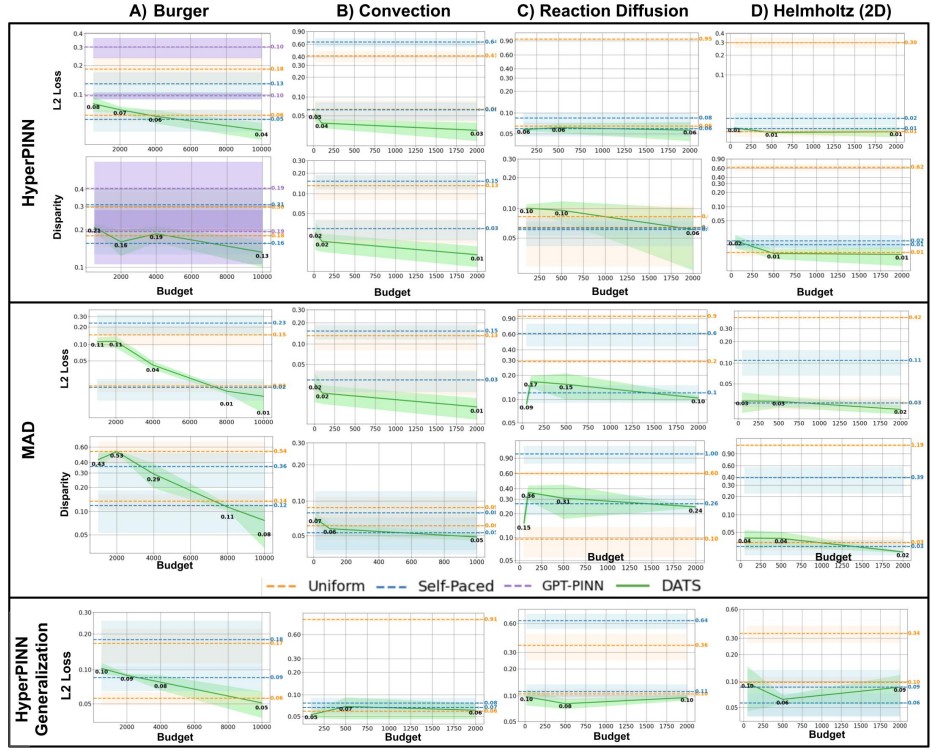

Figure 3: L2 errors and disparity metrics on HyperPINN and MAD-PINN comparing DATS at different residual budgets with baselines obtained at the lower and higher ends of the budget spectrum.

budget per task, and the higher- and lower-bound of PINN errors obtained by the baselines were established at the two ends of the budget spectrum. All results consistently confirmed our hypotheses. First, at the same budgets (the ends of the budget spectrums), DATS consistently out-performed all the baselines, including GPT-PINN on Burger, with a more significant margin of improvements at the lower end of the budgets. Second, DATS was able to reach the lower bound of both the L2 errors and disparity measures of the two baselines (obtained using a budget = 10,000 residual points per task) using only a fraction of their budgets, *i.e.*, ranging from 40% for HyperPINN on the Burgers' equation, to less than 10% on Helmholtz, and to less than 1% on the convection equation comparing to uniform baseline's budget. A similar trend is observed in the results for varying initial conditions on the convection equation, as summarized in Figure .D.7 in Appendix D.7 .

Figure 4A provides examples of DATS and GPT-PINN solutions: while GPT-PINN's ability to recognize harder tasks allows a good performance on harder Burger parameters (*e.g.*, $\lambda = 0.001$), this seemed to be at the expense of the performance on easier Burger parameters (*e.g.*, $\lambda = 0.1$). In comparison, DATS was able to balance its performance across difficult and easy tasks. Figure 5 provides additional visual examples of the PDE solutions obtained with uniform and DATS task sampling at selected residual budgets. More examples can be found in AppendixD.6.

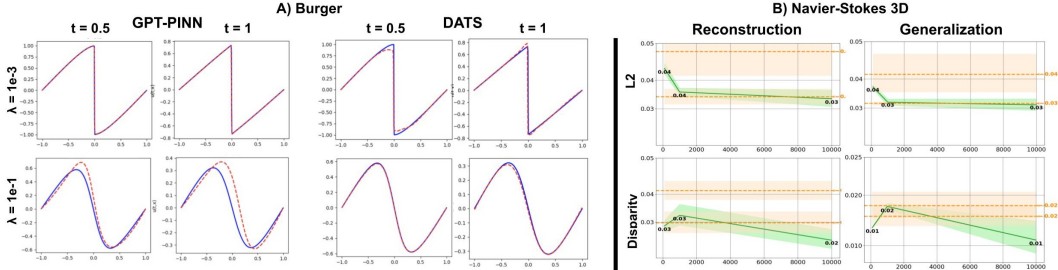

Figure 4: A) Examples of PDE solutions obtained by GPT-PINN and DATS on the Burger equation. B) L2 error and disparity metrics of DATS vs. uniform task sampling on 3D Navier-Stokes equation.

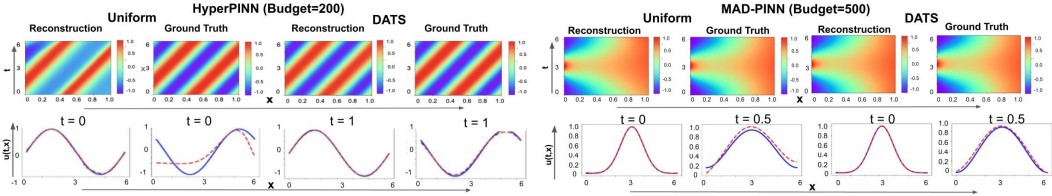

Figure 5: Visual examples of PDE solutions obtained by DATS and uniform task sampling on the convection (left) and reaction diffusion equation (right).

Figure 4B summarizes the results of DATS *vs.* uniform task sampling on HyperPINN on the more complex 3D Navier-Stokes equation (17) with the PDE parameter $a \in [0.5, 1.5]$. Similarly, for PDE configurations included in meta-training, DATS was able to use $40 - 60\%$ of the budget to achieve the performance uniform task sampling achieved using a budget of 10,000 residual points per task, and only $20 - 40\%$ of the budget for generalizing to PDE configurations unseen in meta-training. More details about this experimental setup are included in Appendix C,D.5.

**Results on Generalizing to New PDEs:** The third row of Figure 3 shows L2 errors in PDE solutions when HyperPINN was applied to parameters outside the meta-training sets. The trend is similar to that observed in meta-trained PDE parameters, with notable improvements in some equations like Helmholtz. Detailed results on the disparity metric are in Appendix D.3.

**Computational Cost:** In practice, DATS update of $p^*(\lambda_i)$ in equation 18 can be computed every few iterations: through experiments we found that a period of 200 could potentially outperform a period of 1 while being more computationally efficient (more details presented in Appendix D). With this, the computational overhead by DATS was negligible: on the same computing environments, in HyperPINN, DATS was $3\%$ slower than uniform and $1\%$ slower than self-paced baselines; in MAD-PINN, DATS was $8\%$ and $6\%$ slower than uniform and self-paced baselines, respectively.

## 6 LIMITATIONS AND FUTURE WORK

In this paper, we introduce DATS as the first adaptive task sampler for meta-learning PINNs. By optimizing a training task sampling probability that minimizes the meta-model's validation performance, DATS prevents using unnecessary resources (residual points) for learning easier tasks while improving the performance in learning more difficult PDE configurations. While deriving analytical solutions for the discrete setting commonly used in meta-PINNs, future works need to theoretically examine the convergence of the algorithms presented as well as investigate the ability of DATS to optimize over the continuous distribution of $\lambda$. Furthermore, the evaluation of DATS can be conducted on a broader variety of more complex and higher-dimensional PDEs, meta-PINN models, and PDE configurations. Tuning the regularization hyperparameters in DATS could be challenging and deserves further examinations. Finally, this work considers DATS only in the setting of PINNs, although its underlying concept may be generalizable to operator learning which is known to require a substantial number of training samples of paired PDE configurations and solutions.

ACKNOWLEDGMENTS

This study is supported by the National Key Research and Development Program of China(No: 2020AAA0109502), the Talent Program of Zhejiang Province (No: 2021R51004), the NIH NHLBI grant R01HL145590, the NIH National Institute of Nursing Research (NINR) grant R01NR01830 and NSF OAC-2212548.

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

## A   DATS DISCRETE APPROXIMATION

Below we derive Equation (13) from Equation (12) of the main paper using Karush–Kuhn–Tucker (KKT) first-order necessary condition for an optimal solution. Note again the objective function in Equation (12) in the main paper is:

$$\mathcal{L}(p(\lambda_i)) = \frac{1}{n} \sum_{i=1}^{n} p(\lambda_i) w_i - \beta \, \mathrm{KL}(p(\lambda)||r(\lambda)), \quad w_i = \langle g_{tr,\lambda_i}, \sum_{j=1}^{n} g_{val,\lambda_j}(\theta^t) \rangle \quad (12)$$

Now, differentiating Equation equation 12 with respect to $p(\lambda_i)$ and setting the derivative to zero we derive:

$$\frac{\partial}{\partial p(\lambda_i)} \left( \frac{1}{n} \sum_{i=1}^{n} p(\lambda_i) w_i - \beta \, \mathrm{KL}(p(\lambda)||r(\lambda)) \right) = 0 \quad (13)$$

$$\frac{1}{n} w_i - \beta \log\left( \frac{p(\lambda_i)}{r(\lambda_i)} \right) = 0 \quad (14)$$

$$\log\left( \frac{p(\lambda_i)}{r(\lambda_i)} \right) = \frac{1}{n\beta} w_i \quad (15)$$

$$\frac{p(\lambda_i)}{r(\lambda_i)} = exp(\frac{1}{n\beta} w_i) \quad (16)$$

Re-arranging Equations (4) for deriving $p(\lambda_i)$ and substituting $\beta = n\beta$ we derive at:

$$p(\lambda_i) = r(\lambda_i) * exp(\frac{1}{\beta} w_i) \quad (17)$$

Finally, normalizing $p(\lambda_i)$ to form the probabilities and replacing $r(\lambda_i)$ with $r_i$ we derive at Equation (13).

$$p^*(\lambda_i) = r_i * exp(\frac{1}{\beta} w_i)/ \sum_i r_i * exp(\frac{1}{\beta} w_i) \quad (18)$$

where $r_i = \frac{1}{n}$ can be used for uniform KL regularization, and $r_i = p(\lambda_i)^t$ for consecutive KL regularization.

## B   IMPLEMENTATION DETAILS

In this section, we discuss the specific hyper-parameters on each experiment across all models. Experiments were run on NVIDIA Tesla T4s with 16 GB memory. More information and details can be seen in the provided implementation.

### B.1 BURGERS' EQUATION

**HyperPINN**

- PINN
    - Fully Connected Layers
    - Number of Layers: 7
    - Hidden layers dimenstion: 8
- Hypernet
    - Fully Connected Layers
    - Number of Layers: 7
    - Hidden layers dimenstion: 8
    - Optimizer: ADAM
    - Learning rate: 1e-4 with cosine annealing
    - Epochs: 20000

**MAD-PINN**

- Fully Connected Layers
- Number of Layers: 7
- Hidden layers dimenstion: 128
- Latent Dimension: 16
- Latent Loss Regularizer: 0.001
- Optimizer: ADAM
- Learning rate: 1e-3 with cosine annealing
- Epochs: 20000

### B.2 CONVECTION EQUATION

**HyperPINN**

- PINN
    - Fully Connected Layers
    - Number of Layers: 5
    - Hidden layers dimenstion: 16
- Hypernet
    - Fully Connected Layers
    - Number of Layers: 5
    - Hidden layers dimenstion: 32
    - Optimizer: ADAM
    - Learning rate: 0.001
    - Epochs: 10000

**MAD-PINN**

- Fully Connected Layers
- Number of Layers: 5
- Hidden layers dimenstion: 16
- Latent Dimension: 8
- Latent Loss Regularizer: 0.001
- Optimizer: Adam
- Learning rate: 0.001
- Epochs: 10000

### B.3    REACTION DIFFUSION EQUATION

**HyperPINN**

- PINN
  - Fully Connected Layers
  - Number of Layers: 5
  - Hidden layers dimenstion: 16
- Hypernet
  - Fully Connected Layers
  - Number of Layers: 5
  - Hidden layers dimenstion: 32
  - Optimizer: Adam
  - Learning rate: 0.001
  - Epochs: 10000

**MAD-PINN**

- Fully Connected Layers
- Number of Layers: 5
- Hidden layers dimenstion: 16
- Latent Dimension: 8
- Latent Loss Regularizer: 0.001
- Optimizer: Adam
- Learning rate: 0.001
- Epochs: 10000

### B.4    HELMHOLTZ (2D) EQUATION

**HyperPINN**

- PINN
  - Fully Connected Layers
  - Number of Layers: 5
  - Hidden layers dimenstion: 16
- Hypernet
  - Fully Connected Layers
  - Number of Layers: 5
  - Hidden layers dimenstion: 32
  - Optimizer: Adam
  - Learning rate: 0.001
  - Epochs: 10000

**MAD-PINN**

- Fully Connected Layers
- Number of Layers: 5
- Hidden layers dimenstion: 16
- Latent Dimension: 8
- Latent Loss Regularizer: 0.001
- Optimizer: Adam
- Learning rate: 0.001
- Epochs: 10000

| | L2 | | Disparity | |
|--------|--------|-----------|--------|-----------|
| Period | DATS | Self-Paced | DATS | Self-Paced |
| 1 | 0.0527 | 0.0678 | 0.1317 | 0.1648 |
| 200 | 0.0385 | 0.0699 | 0.0891 | 0.2049 |
| 2000 | 0.120 | 0.0707 | 0.2170 | 0.2048 |

Table D.3: Investigating the effect of the period of updating task probabilities.

## C  NAVIER-STOKES EQUATION

We tested DATS on incompressible Navier-Stokes flow, specifically the three-dimensional Beltrami flow that adheres to the unsteady, incompressible three-dimensional Navier-Stokes equations outlined below:

$$\frac{\partial \mathbf{u}}{\partial t} + (\mathbf{u} \cdot \nabla \mathbf{u})\mathbf{u} = -\nabla p + \frac{1}{Re}\nabla^2 \mathbf{u} \quad in \ \Omega$$
$$\nabla \cdot \mathbf{u} = 0 \quad in \ \Omega$$
$$\mathbf{u} = \mathbf{u}_\Gamma \quad on \ \Gamma_D \tag{19}$$
$$\frac{\partial \mathbf{u}}{\partial n} = 0 \quad on \ \Gamma_N$$
$$\mathbf{u}(\mathbf{x}, 0) = h(\mathbf{x}) \quad in \ \Omega$$

where $t$ is the non-dimensional time, $\mathbf{u}(\mathbf{x}, t) = [u, v, w]^T$ is the non-dimensional velocity vector, $p$ is the non-dimensional pressure, $Re$ is the Reynolds number, $h(\mathbf{x})n$ is the initial condition and $\Gamma_D$ and $\Gamma_N$ denotes the Dirichlet and Neumann boundary conditions, respectively. In Beltrami flow, $Re$=1 and there is the following analytic solution:

$$u(x, y, z, t) = -a[e^{ax}sin(ay + dz) + e^{az}cos(ax + dy)]e^{-d^2 t}$$
$$v(x, y, z, t) = -a[e^{ay}sin(az + dx) + e^{ax}cos(ay + dz)]e^{-d^2 t}$$
$$w(x, y, z, t) = -a[e^{az}sin(ax + dy) + e^{ay}cos(az + dx)]e^{-d^2 t}$$
$$p(x, y, z, t) = -\frac{1}{2}a^2[e^{2ax} + e^{2ay} + e^{2az} + 2sin(ax + dy)cos(az + dy)e^{a(y+z)} \tag{20}$$
$$2sin(ay + dz)cos(ax + dy)e^{a(z+x)}$$
$$2sin(az + dx)cos(ay + dz)e^{a(x+y)}]e^{-2d^2 t}$$

In our experiment, we consider $\mathbf{x} = [x, y, z] \in [-1, 1] \times [-1, 1] \times [-1, 1]$, $t \in [0, 1]$ and d=1 and select $a$ as the PDE parameter in the range$[0.5, 1.5]$.

## D  ADDITIONAL EXPERIMENTS

### D.1  HYPER-PARAMETERS

To decide the period of updating DATS, we conducted a grid search for period $\in \{1, 200, 2000\}$. For a fair comparison with self-paced learning baseline, we also investigated the effect of period of updates on the performance of this baseline as well. As shown in table .D.3 The results indicate that models at period=200 for DATS and period=1 for self-paced learning baseline, were top performing. Note that this experiment is conducted on Burgers equation considering six values of $\lambda \in (1e-3, 1e-1)$ different than the subset of PDE parameters used for the experiments of the main paper.

### D.2  FULL RESULTS ACROSS ALL PDES

Table. D.4 and  D.5 explain the full set of results from figure 3 of section 5.2 on HyperPINN and Mad-PINN.

| HyperPINN | | | | | | |
|---|---|---|---|---|---|---|
| | **Burger** | | | **Convection** | | |
| Method | Budget | L2 | Disparity | Budget | L2 | Disparity |
| Uni. | 500 | 0.184± 0.071 | 0.296±0.106 | 100 | 1.107 ± 0.0177 | 0.064 ± 0.048 |
| S-P | 500 | 0.130 ± 0.040 | 0.308±0.123 | 100 | 0.085 ± 0.018 | **0.063 ±- 0.012** |
| GPT | 500 | 0.301 ± 0.063 | 0.405 ± 0.210 | - | - | - |
| DATS | 500 | **0.079± 0.017** | **0.209 ± 0.0180** | 100 | **0.057 +0.002** | 0.100 ± 0.011 |
| DATS | 2000 | 0.068 ± 0.006 | 0.161±0.045 | 500 | 0.061 ±0.012 | 0.096 ± 0.026 |
| DATS | 4000 | 0.057 ± 0.012 | 0.186±0.032 | - | - | - |
| DATS | 10000 | **0.039 ± 0.010** | **0.132±0.039** | 2000 | **0.056 ±0.021** | **0.061 ± 0.047** |
| GPT | 10000 | 0.097 ± 0.008 | 0.193 ± 0.087 | - | - | - |
| S-P | 10000 | 0.052 ± 0.018 | 0.156±0.035 | 2000 | 0.059 ± 0.016 | 0.061 ± 0.024 |
| Uni. | 10000 | 0.058 ± 0.011 | 0.179+0.039 | 2000 | 0.064 ± 0.008 | 0.082 ± 0.029 |
| | **Reaction-Diffusion** | | | **Helmholtz (2D)** | | |
| Method | Budget | L2 | Disparity | Budget | L2 | Disparity |
| Uni. | 20 | 0.410 ± 0.088 | 0.131 ±0.060 | 50 | 0.298 ± 0.056 | 0.624 ±0.110 |
| S-P | 20 | 0.642 ± 0.103 | 0.153 ± 0.043 | 50 | 0.0169 ± 0.005 | **0.017 ± 0.007** |
| DATS | 20 | **0.052 ± 0.015** | **0.025 ±0.014** | 50 | **0.0108 ± 0.001** | **0.017 ± 0.007** |
| DATS | 100 | 0.037 ± 0.008 | 0.0180 ±0.007 | 500 | 0.0086 ± 0.001 | 0.008 ± 0.003 |
| DATS | 2000 | **0.028 ± 0.012** | **0.011 ± 0.005** | 2000 | **0.009 +0.003** | **0.008 ± 0.005** |
| S-P | 2000 | 0.062 ± 0.024 | 0.029 ±0.012 | 2000 | 0.0104 ± 0.002 | 0.013 ± 0.003 |
| Uni. | 2000 | 0.063 ± 0.029 | 0.029 ±0.015 | 2000 | **0.009 ± 0.002** | 0.009 ± 0.002 |

Table D.4: DATS vs. Uniform (Uni.) vs. Self-Pace (S-P) vs. GPT-PINN (GPT) from figure 3 of section 5.2 on HyperPINN. For readability, the double horizontal line in the Table divides the group of methods using the same budget of residual sampling points. Within the group of the lowest and highest budgets, the best results are bolded among DATS and its baselines. Within the group of DATS with varying intermediate budgets, we underscore the performance better than the best baseline performance achieved at the highest budget.

| MAD-PINN | | | | | | |
|---|---|---|---|---|---|---|
| | **Burger** | | | **Convection** | | |
| Method | Budget | L2 | Disparity | Budget | L2 | Disparity |
| Uni. | 1000 | 0.147± 0.066 | 0.535±0.186 | 100 | 1.284 ± 0.089 | **0.053 ± 0.088** |
| S-P | 1000 | 0.234± 0.110 | **0.356±0.191** | 100 | 0.376 ± 0.084 | 0.665 ± 0.099 |
| DATS | 1000 | **0.111± 0.025** | 0.429 ± 0.102 | 100 | **0.167 ± 0.032** | 0.361 ± 0.088 |
| DATS | 2000 | 0.113 ± 0.031 | 0.531±0.024 | 500 | 0.155 ± 0.067 | 0.308 ± 0.165 |
| DATS | 4000 | 0.042 ± 0.009 | 0.290±0.114 | - | - | - |
| DATS | 8000 | 0.013 ± 0.002 | 0.114±0.114 | - | - | - |
| DATS | 10000 | **0.010 ± 0.007** | **0.077±0.053** | 2000 | **0.105±0.009** | **0.241±0.007** |
| S-P | 10000 | 0.015 ± 0.009 | 0.119±0.080 | 2000 | 0.121 ± 0.036 | 0.264±0.086 |
| Uni. | 10000 | 0.016 ± 0.004 | 0.135+0.036 | 2000 | 0.290 ± 0.013 | 0.602± 0.041 |
| | **Reaction-Diffusion** | | | **Helmholtz (2D)** | | |
| Method | Budget | L2 | Disparity | Budget | L2 | Disparity |
| Uni. | 10 | 0.931 ± 0.004 | 0.088 ± 0.025 | 50 | 0.419±0.027 | 1.193±0.067 |
| S-P | 10 | 0.953 ± 0.067 | 0.080 ± 0.051 | 50 | 0.109±0.052 | **0.039±0.209** |
| DATS | 10 | **0.051 ± 0.007** | **0.071 ± 0.011** | 50 | **0.027±0.011** | 0.040±0.018 |
| DATS | 100 | 0.045 ± 0.020 | 0.058 ± 0.008 | 500 | 0.027±0.006 | 0.040±0.014 |
| DATS | 1000 | **0.035 ± 0.002** | **0.049 ± 0.004** | 2000 | **0.020±0.004** | **0.022±0.001** |
| S-P | 1000 | 0.053 ± 0.019 | 0.053 ± 0.023 | 2000 | 0.025±0.004 | 0.028±0.010 |
| Uni. | 1000 | 0.040 ± 0.012 | 0.061 ± 0.018 | 2000 | 0.026±0.007 | 0.033±0.011 |

Table D.5: DATS vs. Uniform (Uni.) vs. Self-Pace (S-P) vs. GPT-PINN (GPT) from figure 3 of section 5.2 on MAD-PINN.

## D.3 RESULTS ON GENERALIZING TO NEW PDES

| HyperPINN Generalization | | | | | | |
|---|---|---|---|---|---|---|
| **Burger** | | | | **Convection** | | |
| **Method** | Budget | L2 | Disparity | Budget | L2 | Disparity |
| Uni. | 500 | 0.167 ± 0.065 | **0.025 ± 0.094** | 100 | 0.907 ± 0.047 | 0.080 ± 0.063 |
| S-P | 500 | 0.180 ± 0.092 | 0.278 ± 0.098 | 100 | 0.077 ± 0.016 | 0.034 ± 0.012 |
| DATS | 500 | **0.102 ± 0.014** | 0.160 ± 0.009 | 100 | **0.053 ± 0.006** | **0.022 ± 0.012** |
| DATS | 2000 | 0.089 ± 0.003 | 0.135 ± 0.020 | 500 | 0.069 ± 0.028 | 0.034 ± 0.004 |
| DATS | 4000 | 0.077 ± 0.014 | 0.137 ± 0.026 | - | - | **-** |
| DATS | 10000 | **0.051 ± 0.016** | **0.084 ± 0.031** | 2000 | 0.061 ± 0.018 | **0.022 ± 0.004** |
| S-P | 10000 | 0.085 ± 0.024 | 0.129 ± 0.051 | 2000 | 0.067 ± 0.015 | 0.049 ± 0.012 |
| Uni. | 10000 | 0.056 ± 0.009 | 0.114 ± 0.016 | 2000 | **0.059 ± 0.010** | 0.028 ± 0.014 |
| **Reaction-Diffusion** | | | | **Helmholtz (2D)** | | |
| **Method** | Budget | L2 | Disparity | Budget | L2 | Disparity |
| Uni. | 20 | 0.357 ± 0.133 | 0.244 ± 0.061 | 50 | 0.336 ± 0.081 | 0.440 ± 0.091 |
| S-P | 20 | 0.641 ± 0.131 | 0.504 ± 0.448 | 50 | **0.087 ± 0.059** | 0.080 ± 0.049 |
| DATS | 20 | **0.098 ± 0.024** | **0.095 ± 0.018** | 50 | 0.096 ± 0.062 | **0.010 ± 0.089** |
| DATS | 100 | 0.081 ± 0.013 | 0.102 ± 0.022 | 500 | 0.062 ± 0.009 | 0.050 ± 0.026 |
| DATS | 2000 | **0.095 ± 0.008** | 0.123 ± 0.015 | 2000 | 0.086 ± 0.039 | 0.073 ± 0.075 |
| S-P | 2000 | 0.112 ± 0.027 | 0.129 ± 0.031 | 2000 | **0.057 ± 0.020** | **0.030 ± 0.021** |
| Uni. | 2000 | 0.105 ± 0.018 | **0.114 ± 0.008** | 2000 | 0.098 ± 0.018 | 0.101 ± 0.019 |

Table D.6: DATS vs. Uniform (Uni.) vs Self-Pace (S-P) generalization results for HyperPINN across all PDEs.

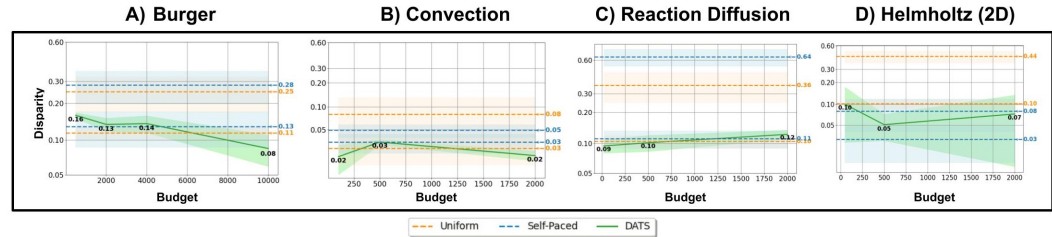

Figure D.6: HyperPINN generalization results (disparity) on PDE parameters not included in meta-training.

Figure.D.6 summarizes the disparity metric when HyperPINN is applied to PDE parameters outside meta-training sets (generalization) and table.D.6 includes the full set of results presented in the third row of figure. 3

## D.4 Results on Generalizing to New Initial Conditions

Figure. D.7 compares DATS with uniform and self-pace baselines on MAD-PINN when tested on initial conditions unseen during training.

## D.5 Results on 3D Navier-Stokes equation

Table. D.7 presents the full set of results including L2 and disparity metrics using DATS and uniform task-sampling baselines for solving 3D Navier-Stokes equation with HyperPINN and parameter configuration $a \in [0.5, 1.5]$. A subset of 6 parameters is used for training and 4 unseen parameters during the training are used for generalization experiments.

## D.6 Example solutions

**Loss convergence & examples of visual results:** Figures. D.8 showcases validation $L2$ error comparison of DATS with Uniform and self-paced learning baselines using HyperPINN during training for the example budget of $b = 500$ on Burgers' equation. The thirds column in this figure shows how the probabilities assigned to each task (pde parameter instance) changes during training. In

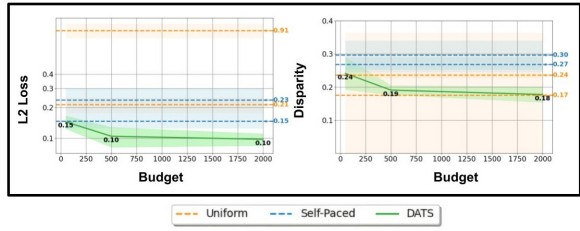

Figure D.7: MAD-PINN results on learning across initial conditions for the convection equation.

| 3D Navier-Stokes | | | | | |
|---|---|---|---|---|---|
| Method | Budget | Reconstruction | | Generalization | |
| | | L2 | Disparity | L2 | Disparity |
| Uniform | 100 | $0.048 \pm 0.008$ | $0.041 \pm 0.003$ | $0.041 \pm 0.007$ | $0.018 \pm 0.003$ |
| DATS | 100 | $\mathbf{0.043 \pm 0.002}$ | $\mathbf{0.028 \pm 0.003}$ | $\mathbf{0.037 \pm 0.001}$ | $\mathbf{0.014 \pm 0.001}$ |
| DATS | 1000 | $0.036 \pm 0.002$ | $0.033 \pm 0.005$ | $\underline{0.032 \pm 0.001}$ | $0.018 \pm 0.001$ |
| DATS | 10000 | $\mathbf{0.034 \pm 0.004}$ | $\mathbf{0.021 + 0.007}$ | $\mathbf{0.031 \pm 0.003}$ | $\mathbf{0.012 \pm 0.005}$ |
| Uniform | 10000 | $0.035 \pm 0.003$ | $0.03 + 0.005$ | $0.032 \pm 0.001$ | $0.017 \pm 0.002$ |

Table D.7: DATS vs Uniform task sampling for reconstruction and generalization of 3D Navier-Stokes equation with HyperPINN.

this experiment, DATS probabilities are updated every 200 epochs. Similarly, Figure . D.9 compares MAD-PINN results at the example budget of $b = 1000$. Figures. D.10, D.11,D.12 and D.13 visualize example PDE solutions of convection, reaction diffusion, Helmholtz (2D) accordingly. Figure. D.13 compares MAD-PINN before and after applying DATS on new initial conditions for the convection equation.

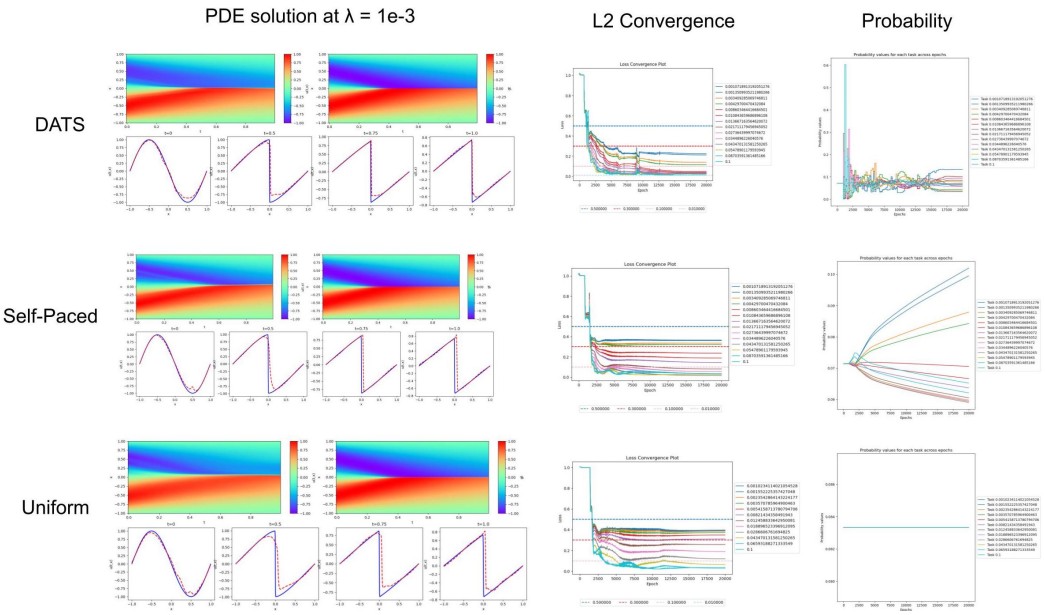

Figure D.8: Comparison of PDE solution at $\lambda = 1e - 3$, L2 loss convergence, and probabilities at 500 residual points budget using HyperPINN.

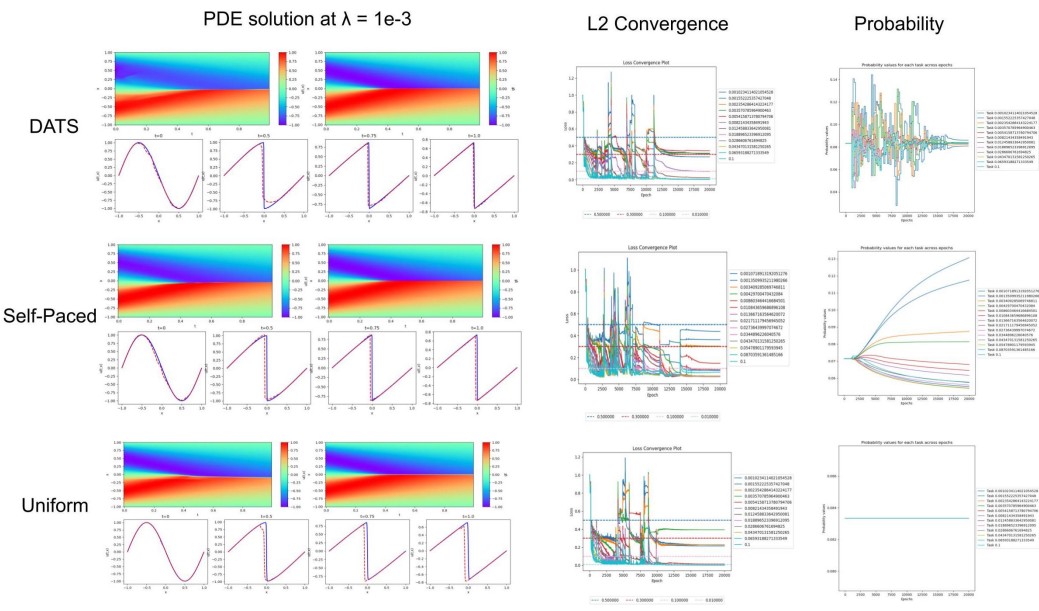

Figure D.9: Comparison of PDE solution at $\lambda = 1e - 3$, L2 loss convergence, and probabilities at 1000 residual points budget using MAD-PINN.

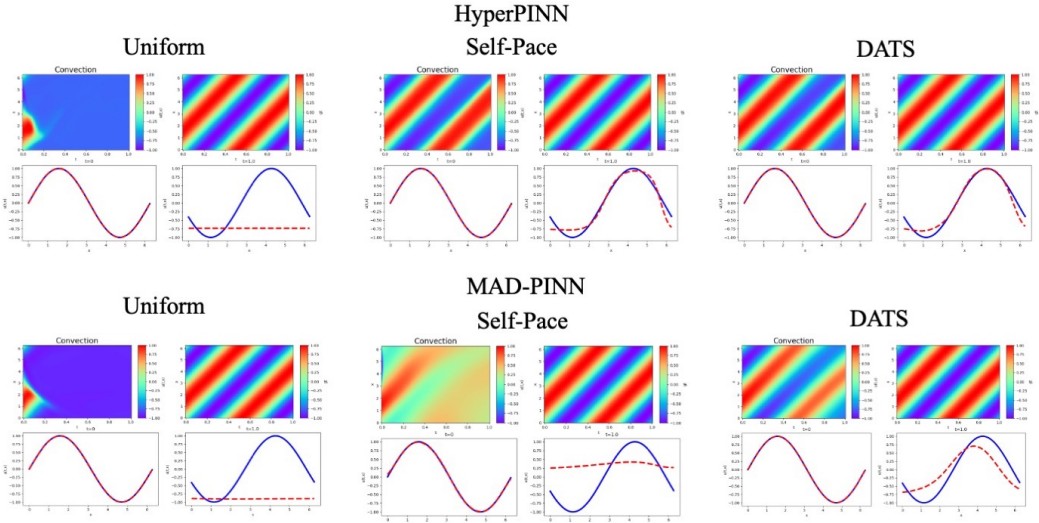

Figure D.10: Comparison of Convection equation solution at $\beta = 9$ and 100 residual points budget.

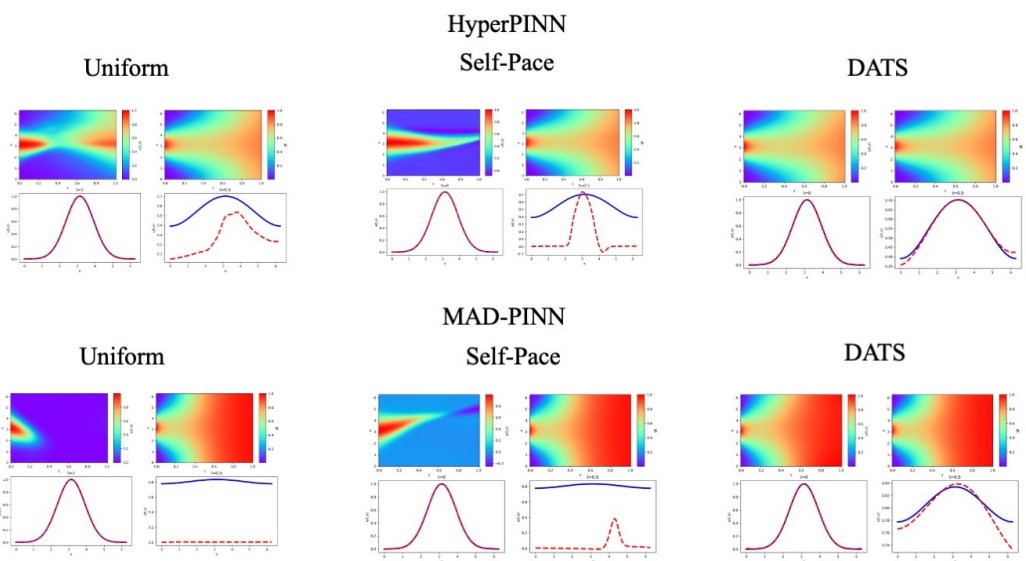

Figure D.11: Comparison of Reaction diffusion equation solution at $\nu, a = 2.5, 2.5$ and 20 residual points budget.

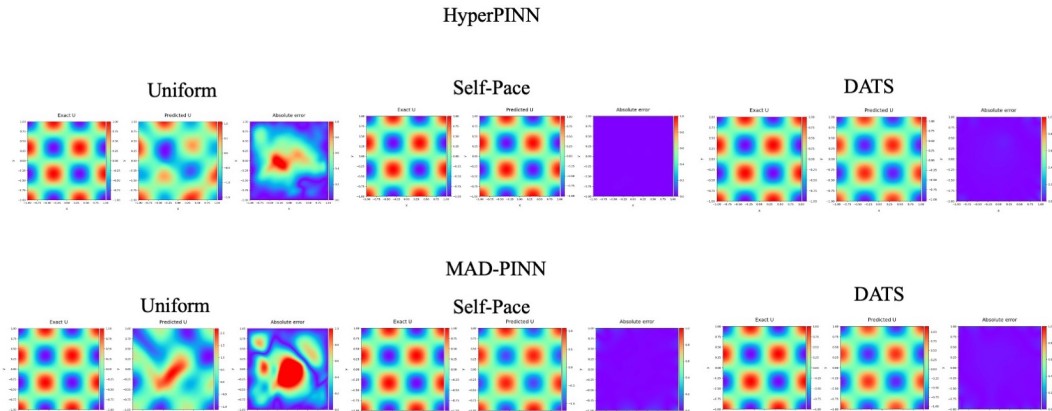

Figure D.12: Comparison of Helmholtz equation solution at $a_1, a_2 = 1.5, 1.5$ and 50 residual points budget.

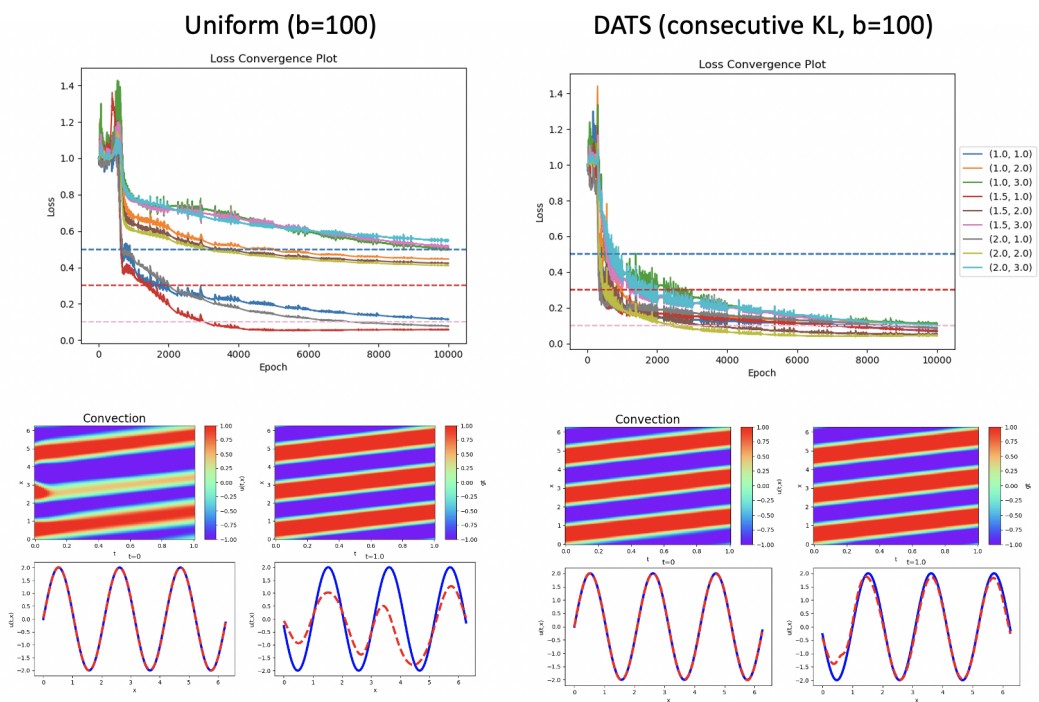

Figure D.13: Comparing MAD-PINN and DATS on new initial condition $u(x, 0) = 2sin(3x)$. Top row indicates L2 loss and bottom row compares the results on $\beta = 1$ (most difficult task in training).

