# OpenReview forum: "DATS: Difficulty-Aware Task Sampler for Meta-Learning Physics-Informed Neural Networks"
_ICLR.cc/2024/Conference — ICLR 2024 poster_

### Official Review · Reviewer_LK9J · 2023-10-29

**Soundness:** 3 good
**Presentation:** 2 fair
**Contribution:** 3 good
**Rating:** 6
**Confidence:** 3

**Summary:**

This paper presents a sampling strategy for meta-learning in physics-informed neural networks (PINNs). The key idea is to conduct sampling based on the difficulty. The authors provide an analytical solution to optimize the sampling probability, with a regularization term. Experiments show improved performance over uniform sampling. An ablation study has been presented to help understand the method. The method also shows better performance under the same budget.

**Strengths:**

1. Meta-learning for PINN is a promising solution to generalize PINNs so that we do not have to train from scratch for any new PDE.
2. The proposed sampling strategy is general and can be combined with the existing meta-learning strategy.
3. Experiments show the proposed method outperforms the uniform sampling.

**Weaknesses:**

1. It is necessary to report training costs in terms of running time. Although the method can improve the sampling efficiency, the sampling strategy itself could be more time-consuming than uniform sampling. It is unclear whether the method can bring actual benefits in training time compared with the baseline.
2. The method is only tested on three benchmarks. Results on more benchmarks are encouraged to test the generalizability of the proposed method, such as Heat, Wave, and Advection.

**Questions:**

What is the actual training time improvement?

---

> ### Author Response · Authors · 2023-11-22
> **Response to Reviewer LK9J**
>
> Dear Reviewer LK9J,
>
> Thank you for your constructive feedback on our manuscript. We are pleased to address the concerns you have raised regarding our proposed sampling strategy for meta-learning in physics-informed neural networks (PINNs).
>
> 1. **Training Cost Reporting and Actual Training Time Improvement**:
> We acknowledge the importance of evaluating the computational efficiency of our method alongside its performance benefits. That is the reason we included the  subsection "computational cost" in section 5.3. However, it is important for us to clarify that the motivation for DATS is not to improve the sampling “efficiency” in terms of computational cost of the meta-learning. Instead, the goal is to improve the sampling “efficacy” for meta-learning PINNs to improve the accuracy of the PINNs both for PDE configurations included in the meta-learning as well as for those outside the meta-training distributions (improving generalization).  Our findings showed that, when using the same budget for residual points, DATS improved the PINNs’ performance significantly with only a modest overhead to determine task difficulty ( a 3-8% increase in the overall training time). More importantly, to achieve the same PINN performance obtained by uniform sampling, DATS can reduce the number of residual points needed to a range from less than 1% to 40% of what is needed using uniform sampling, effectively reducing computational burden.
>
> 2. **Benchmarks**:
> While we acknowledge the potential benefits of expanding the dataset, we would like to  clarify that we have conducted experiments on **four**benchmark datasets including "Burgers", "Convection", "Reaction Diffusion" and "Helmholtz" that are commonly used in PINN literature: this selection ofPDEs is on par with those published [1,2]. In addition, we have added another challenging PDE, the 3D Navier-Stokes equation. Please see details in bullet one of Overall Response for the added results and disucssion.
>
> 1:[Huang, Xiang, et al. "Meta-auto-decoder for solving parametric partial differential equations." Advances in Neural Information Processing Systems 35 (2022): 23426-23438.]
>
> 2:[Kim, Jungeun, et al. "DPM: A novel training method for physics-informed neural networks in extrapolation." Proceedings of the AAAI Conference on Artificial Intelligence. Vol. 35. No. 9. 2021.]

---

### Official Review · Reviewer_USEY · 2023-10-31

**Soundness:** 3 good
**Presentation:** 3 good
**Contribution:** 3 good
**Rating:** 6
**Confidence:** 4

**Summary:**

The paper introduces a novel approach to Physics-Informed Neural Networks (PINNs) with a focus on unsupervised learning for parameterized Partial Differential Equations (PDEs). The authors propose optimizing the probability distribution of task samples to minimize error during meta-validation. They transform the problem theoretically into a discretized form suitable for optimization and introduce two optimization strategies: optimizing the residual points sampled and the loss weight across different tasks. The paper presents experiments on several equations, showcasing improvements over baseline methods.

**Strengths:**

1. Clarity and Presentation: The paper is well-written, making the novel method and its distinctions from baselines clear. The method is explained in a step-by-step manner, which is easy to follow.
2. Innovative Approach: The meta-learning method for PINNs is a fresh take on solving parameterized PDEs, and it is well grounded in theory.
3. Comprehensive Experiments: The authors conduct experiments on various equations, providing a thorough evaluation of their method.
4. Effective Visualization: The results are presented in a clear manner, with visualizations that aid in understanding the improvements made.

**Weaknesses:**

1. Mischaracterization of Data Loss: In Section 3, the paper inaccurately defines data loss as the loss of boundary conditions. This is a mischaracterization, especially for Neumann or Robin boundary conditions, which only penalize normal derivatives rather than resulting in data loss.

2. Formatting and Clarity of Figures: Some figures in the paper could be improved for better clarity and understanding. For instance:
(1) Figures 3 and 4 would benefit from added grids and key values annotated directly on the figures.
(2) The scales in Figure 5 are too small to read, making it difficult to interpret the results.
The authors should review and adjust these figures to enhance clarity.

3. Lack of Comparison with State-of-the-Art: The paper could be strengthened by including a comparison with state-of-the-art methods in the field, providing a clearer context of the method's performance.

4. Limited Discussion on Limitations: The paper does not adequately discuss the limitations of the proposed method, which is crucial for readers to understand the potential challenges and boundaries of the approach.

5. Potential Overfitting: Given the nature of the meta-learning approach, there could be a risk of overfitting to the tasks at hand. The paper could benefit from a discussion on how this risk is mitigated or how the method performs under such circumstances.

**Questions:**

1. Clarify Mischaracterizations: The authors should revisit Section 3 to correct the mischaracterization of data loss and provide a more accurate description.

2. Improve Figure Formatting: Enhancements should be made to the figures to improve readability and clarity, as this will aid in better conveying the results and contributions of the paper.

3. Include Comparison with State-of-the-Art: Adding comparisons with leading methods in the field will provide a clearer benchmark of the proposed method's performance.

4. Discuss Limitations and Potential Overfitting: A section discussing the limitations of the method and addressing potential overfitting concerns would add depth to the paper and provide a more balanced view of the approach.

With these enhancements, the paper would offer a more comprehensive and clear presentation of the proposed method, its strengths, and its potential areas for improvement.

**Details Of Ethics Concerns:**

Not apply.

---

> ### Author Response · Authors · 2023-11-22
> **Response to Reviewer USEY (Part 1)**
>
> Dear Reviewer USEY,
>
> Thank you for your thorough review of our manuscript and for recognizing the strengths of our work. We have addressed each of the points raised in your review to enhance the quality and clarity of our manuscript.
>
> 1.**Clarifications and Corrections in Section 3**:
>
> Thanks for pointing this out. We realize that the term "data loss" in the context of Physics-Informed Neural Networks (PINNs) was not precisely used and may have led to confusion, particularly in relation to boundary conditions.
> In PINNs, "data loss" traditionally refers to the component of the loss function that quantifies the error in the neural network's predictions against known solutions or measurements. Specifically, for PINNs, data loss encompasses the error in satisfying initial conditions, boundary conditions (including Dirichlet, Neumann, or Robin), and any interior constraints.
> For Dirichlet conditions, the data loss is indeed a direct measure of the discrepancy between the network outputs and the true boundary values. However, for Neumann or Robin conditions, which involve derivatives, the term "data loss" should more accurately reflect the error in satisfying the prescribed fluxes or a combination of flux and function values on the boundaries. We acknowledge that our initial description may have oversimplified this aspect by not distinguishing between the types of boundary conditions. In the revised manuscript we removed the term "data loss" and replaced it with "initial and boundary condition loss". Eqs (2-3) are revised accordingly to include different types of boundary conditions.
>
> 2.**Improvements in Figure Formatting**:
>
> We have revised Figures 3 and 4 to include grids, key values and have enlarged the fonts and scales. Furthermore, tables of results (Tables C4, C5, and C6) are added in appendix C to aid with understanding of these figures. Figure 5 is updated with enlarged fonts and scales and all figures in the appendix will be updated accordingly for the camera ready version.

---

> > ### Author Response · Authors · 2023-11-22
> > **Response to Reviewer USEY (Part 2)**
> >
> > 3.**Comparison with State-of-the-Art**:
> > In the presented results, we included two baselines – one represents uniform task sampling as the standard practice in meta-learning PINNs, and one represents self-paced task sampling – note that the latter is not an existing method, but one we extend from existing works that use self-paced strategies for residual point sampling (for training a single PINN) [1] to task sampling (for meta-training multiple PINNs). According to the suggestion by Reviewer Z9PQ, we also added a new baseline that aims to learn multiple PINNs at the same time [2] – please refer to Overall Response bullet 2 for details on the new results.
> > Beyond these, we are not aware of any other states-of-the-art that can be considered relevant baselines. As discussed in the original manuscript, relevant works in the field can be categorized in the following groups:
> > Task Sampling Techniques in General Meta Learning: While adaptive task sampling is an active topic in general meta learning (as those discussed in our Related Works), the distinct attributes of PINNs present challenges such that the direct application of these general methodologies is not trivial. Our work represents a first step in this direction. As far as our knowledge extends, no adaptations of these methods to the meta-PINN context have been explored prior to our work. Despite this, we have extended the self-paced learning baseline to the meta-PINN for the first time in the literature and compared DATS against this baseline. In the context of PINNs, self-paced learning has only been used to select training samples (collocation points), but not for a variety of tasks in terms of PDE configurations for meta-learning PINNs prior to our work.
> > Other PDE Solvers Besides PINNs, such as Operator Learning (e.g., DeepONets): These works are reliant on supervision from explicit PDE solutions while the training of PINN is unsupervised, so they can be considered orthogonal (and some emerging work combining them has been attempted [3]). Because DATS focuses on addressing challenges associated with meta-learning **unsupervised** PINNs for multiple PDE configurations, we consider the comparison of other **supervised** PDE-solving strategies out of the scope of our current work but focused on how DATS improves existing baselines of meta-PINNs ( HyperPINN [4] and MAD-PINN [5]) instead. In the meantime, it is noteworthy that the training of DeepONets, etc, requires a large number of training samples over the space of PDE configurations that are treated uniformly. This suggests a potential extension of DATS to enable adaptive sampling of PDEs for operator learning, an interesting future research direction for us to consider.
> > 1: Gu, Yiqi, Haizhao Yang, and Chao Zhou. "Selectnet: Self-paced learning for high-dimensional partial differential equations." Journal of Computational Physics 441 (2021): 110444.
> > 2: Chen, Yanlai, and Shawn Koohy. "GPT-PINN: Generative Pre-Trained Physics-Informed Neural Networks toward non-intrusive Meta-learning of parametric PDEs." Finite Elements in Analysis and Design 228 (2024): 104047.
> > 3: [Wang, Sifan, Hanwen Wang, and Paris Perdikaris. "Learning the solution operator of parametric partial differential equations with physics-informed DeepONets." Science advances 7.40 (2021): eabi8605.]
> > 4: [de Avila Belbute-Peres, Filipe, Yi-fan Chen, and Fei Sha. "HyperPINN: Learning parameterized differential equations with physics-informed hypernetworks." The symbiosis of deep learning and differential equations. 2021]
> > 5:[Huang, Xiang, et al. "Meta-auto-decoder for solving parametric partial differential equations." Advances in Neural Information Processing Systems 35 (2022): 23426-23438.
> >
> > 4.**Discussion of Limitations Potential Overfitting**:
> >
> > We expanded the Limitation discussion (Section 6) in the original manuscript to include additional comments received from the reviewers.
> > Thanks for pointing out the importance of  generalization.e would like to emphasize that generalization ability is a strong motivation of DATS – note that the DAT’s optimization objective is to find task sampling probability to improve the PINNs’ performance on “validation” residual points. Furthermore, our experiments had a specific focus to assess the generalization capabilities of the method: in Section 5.3 of our paper, we delved into the performance of DATS when applied to new PDE configurations that were not included in the training dataset. Figure 6, in particular, displays the generalization capabilities of our approach by illustrating the L2 errors and disparity metrics for these unseen PDE configurations. The empirical evidence provided in these sections strongly suggests that DATS not only enhances the learning process but also ensures that the meta-model retains its predictive accuracy when faced with novel tasks.

---

### Official Review · Reviewer_Z9PQ · 2023-11-01

**Soundness:** 3 good
**Presentation:** 3 good
**Contribution:** 3 good
**Rating:** 6
**Confidence:** 4

**Summary:**

This paper introduces a difficulty-aware task sampler (DATS) for meta-learning of PINNs. The model takes the variance of the difficulty of solving different PDEs into consideration by optimizing the sampling probability of meta-learning. An analytic approximation of the relationship of meta-model and sampling probability is provided to enhance learning. DATS is shown to improve the overall performance of meta-learning PINNs.

**Strengths:**

1) Quality: The performance of the approach seems good in the empirical part of the paper and the ablation study is detailed.
2) Originality: The proposed two strategies to utilize $p^*$ are interesting and the comparison and analysis are comprehensive.

**Weaknesses:**

The main weakness is the clarity and correctness in both methodology and experiments

1) Lack of intuitive explanation for the math derivations in section 4.1, making it hard to follow.  For example, the $w_i=  \langle g_{tr}, g_{val} \rangle$ in Eq.9 may be intuitively interpreted as "assign the weight according to gradient similarity between train and valid", I guess.

2) There are some misleading typos and unexplained assumptions in section 4.1.
* The LHS of Eq(7) should be $l_{\text {val }, \lambda}\left(\theta^{t+1}\right)$, but not $l_{\text {val }, \lambda}\left(\theta^*\right)$ written in Line 5 of this paragraph.
* And similarly, the LHS of Eq(8) should be $p^{t+1}(\lambda)$ not $p^{*}(\lambda)$.
* In Line4-5 of this paragraph, the gradient descent of training loss is defined as $\theta^{t+1} = \theta^t-\eta \int_\lambda p(\lambda) \nabla_\theta l_{t r, \lambda}\left(\theta^t\right) d \lambda$, which assumes that training loss is defined as in Eq(11), the so-called DATS-w. But  DATS-rp loss in Eq(12) does not follow this assumption, thus the analysis is invalid for it.
* The authors assume that the proposed iterative scheme for $p^*,\theta^*$ converges and that adding regularization further stabilizes the convergence, without explanations. Intuitively, the first-order Taylor expansion is used in Eq(7), thus the step size $\theta$ should be small enough to stabilize it. Additionally, the discrete approximations may also introduce errors.

3) The experimental performance of sampling strategies (Section 5.2 and Appendix C) is not reported clearly. There are only figures in the main text. Fig.3,4,6 are difficult to extract information from since the lines and shades overlap heavily.  And since there are no digital numbers available, it is hard to compare the results. For example, in Fig.C.12, it seems Self-pace has the same performance as DATs.

**Questions:**

1) In Fig.3,4,6,  Why do the uniform and self-paced baselines only have higher and lower bounds of errors, not curves at different residual budgets?

2) How does the DATS compare with the Reduced Basis Method, e.g., [1]?

3) Does DATS also perform well on more difficult PDEs, such as with discontinuity and high-dimension? Examples includes the shock tube of compressible Euler's equation and  2d/3d N-S equation.

Refs:
[1] Chen, Yanlai, and Shawn Koohy. "GPT-PINN: Generative Pre-Trained Physics-Informed Neural Networks toward non-intrusive Meta-learning of parametric PDEs." Finite Elements in Analysis and Design 228 (2024): 104047.

---

> ### Author Response · Authors · 2023-11-22
> **Response to Reviewer Z9PQ (Part1)**
>
> Dear Reviewer Z9PQ,
>
> Thank you for your critical assessment and valuable comments on our manuscript. We have revised the manuscript, taking your feedback into careful consideration, and would like to present the following detailed clarifications and justifications:
>
> **Major clarifications and revisions**
>
> We have made major revisions and clarifications on the following points and included details in the Overall Response to all reviewers. Please refer to the details therein.
>
>    - Additional results on a complex PDE (3D N-S) as suggested.
>    - Additional results on a comparison with the Reduced Basis Method as suggested.
>    - Clarification on the validity of DATA-rp derivation:
>
> **Additional clarifications and revisions**
>
> 1. *Intuitive Explanation for Mathematical Derivations in Section 4.1*
>
> We have attempted to add additional explanations to aid the understanding of the mathematical framework throughout (in blue fonts). For instance, after Eq (9), we have added the following: *Intuitively, w_{i} measures the gradient similarity between the training loss of task \lambda_{i} and the validation loss across all tasks. This results in a higher w_{i} to a PDE configuration \lambda_{i} that is most beneficial for reducing the validation loss across all tasks.*
>
> 2. *The use of iterative optimization*:
>    - Indeed, the optimization objective in Eq (6) describes the overall objective for our formulated problem, while starting with Eqs (7) and onward, we have moved into the derivation of iterative solutions to Eq (6) and thus the introduction of superscript t notating the iterations. We have added this clarification leading to the derivation of Eq (7), and revised all relevant notations in the following sections.
>     - We did not discuss the convergence about the iterative optimization approach used, because gradient-based iterative optimization underlies the training of almost all neural networks including bi-level nested optimization commonly used in meta-learning – from this aspect, our proposed iterative schedule for solving p and \theta did not introduce additional assumptions that deserve special attention different from those established for iterative numerical optimization.
>     - However, we do acknowledge that the first-order Taylor expansion used in Eq (7) is a key piece in our derivation that sets us apart from a typical iterative solution to the objectives in Eq (6), which simplifies the optimization procedure but does introduce an approximation that deserves further expansion in the manuscript . In fact, the use of a single step of gradient descent in the inner-level optimization (of theta_t) to simplify the outer-level optimization (of lambda) via first-order Talor expansion is not an uncommon solution to nested optimization: for instance, it also underpins the first-order MAML (model agnostic meta-learning) algorithms such as REPTILE [1], which already showed that this first-order approximation has the same leading-order terms as the original optimization. We have added this connection and citation to the text following Eq (8).
>       - Indeed for the Taylor Expansion in Eq (7) to hold, the step size in theta need to be small; this step size is mainly controlled by the hyperparameter eta (while the integral term controls the direction).  This fact was also acknowledged in Reptile [1]. Experimentally,  we have conducted extensive empirical evaluations to determine appropriate ranges for η, ensuring that within these ranges, our iterative scheme consistently converges. The main body of experiments are conducted using η = 1e-3 based on these observations.
>        - The incorporation of the Kullback–Leibler (KL) divergence as a regularization term stabilizes the task probability assignments, ensuring that the iterative updates do not lead to erratic changes in p(\lambda). The hyperparameter \beta modulates the strength of this regularization. This balance is important for the stability of the convergence, as it prevents the optimization from becoming overly aggressive or conservative. We have conducted extensive experiments for tuning this hyperparameter. These evaluations are included in Table.1 as part of the ablation study.
>
> [1] Alex Nichol and Joshua Achiam and John Schulman, On First-Order Meta-Learning Algorithms

---

> > ### Author Response · Authors · 2023-11-22
> > **Response to Reviewer Z9PQ (Part2)**
> >
> > 3. **Experimental Performance and Reporting**: We summarized the large amount of results we had (two base meta-PINNs, four PDE equations, and two baselines with DATS at different budgets) in one figure due to the space limitation to incorporate them in tables. In this revision, We have updated Figures 3,4, and 6 to include grids and key values on the figures for more clarity. We have also included the complete table listing quantitative results corresponding to these figures in Appendix C, Table C4, C5 and C6. It is important to note that the goal is DATS is not necessarily to improve PINN performance for each every PDE configuration, but to reduce performance disparity among PDE configurations while improving overall performance. The examples in Figure C12 (now Figure D13 in the revised manuscript) represent a specific PDE configuration in the Helmholtz equation where self-paced and DATS sampling obtained similar solution visually using a budget of 50 residual points per task  – the quantitative results (Table D5, Helmholtz, budget 50) however showed that across all PDE configurations, DATS significantly outperformed self-paced baseline (0.027 \pm 0.011 vs. 0.109 \pm 0.052 for L2 error metrics).
> >
> > We only reported the two baselines (uniform and self-paced) on the high and low-end of the budget spectrum before it is observed that, compared with the two baselines using the high end of the budget (the bottom orange/blue bar for uniform/self-paced baseline),
> > DATS already out-performs them with smaller budgets (either using just the lower-end of the budgets (such as in Fig 3-B Convection functions), or using a budget somewhere in the middle of the spectrum (all the rest of the cases reported except in Fig 3-C for the disparity metrics of MAD-PINN). We consider these to provide compelling evidence that DATS is able to achieve the same performance as the baselines using a much smaller budget.

---

> > > ### Comment · Reviewer_Z9PQ · 2023-11-23
> > >
> > > Thank you for the replies and the additional experiments, I decided to raise my score to 6.

---

### Author Response · Authors · 2023-11-22
**Overall response (part 1)**

In response to the constructive feedback provided by all the reviewers, we have included major revisions and clarification in the revised manuscript as summarized below (main revisions are highlighted in blue texts in the revised manuscript).  Additional revisions are detailed in the individual response to each reviewer.

**New PDE (reviewer Z9PQ and LK9J):** We have included a more complex 3D Navier-Stokes PDE and compared DATS with the baseline of uniform task sampling on HyperPINN. The summary of results is added to Fig 4B and Table D7 in Appendix D5. The same trend as observed from the other PDEs was obtained: for PDE configurations included in meta-training, DATS was able to use 40-60% of the budget to achieve the best test performance uniform task sampling achieved using the full budget, and only 20-40% of the budget to achieve the best uniform task sampling performance in generalizing to PDE configurations unseen in meta-training.

**New baseline (reviewer Z9PQ and USEY):** We have included GPT-PINN as a new baseline on the Burger equation for HyperPINN. GPT-PINN learns to use pre-trained PINNs for a finite set of PDE configurations as activation functions in the PINN network of a new PDE. This selective process in determining when to add a neuron for a given PDE configuration inherently considers the difficulty of a PDE task, and thus relevant to DATS from this aspect. We thus added the discussion of GPT-PINN in the Related Works, and included it as a baseline on the Burger equation. The results were added to Fig 3A and Fig 4A (as well as Table D4 in Appendix D2). Fig 3A showed that DATS, even when using the low end of the budget spectrum, outperformed the best performance of GPT-PINN obtained using the maximum budget. Fig 4A gives visual examples of why: while while GPT-PINN’s ability to recognize harder tasks allows a good performance on harder Burger parameters (e.g., \lambda = 0.001), this seemed to be at the expense of the performance on easier Burger parameters (e.g.,\lambda  = 0.1). In comparison, DATS was able to balance its performance across difficult and easy tasks.

We stress again – as noted in our Introduction and Related Works sections –  that there is a lack of relevant baselines to be included in this work because DATS represents the first adaptive task-sampling approaches to meta-PINNs. While related concepts exist in general meta-learning, they are not trivial to directly extend to the PINN due to its unique “unsupervised” learning nature. In parallel, adaptive sampling strategies are being studied in PINN literature but at the level of sampling "residual points" for training a single PINN. In this work, we not only included the uniform task sampling strategy used in standard meta-PINN as a baseline, but also added a self-paced task sampling strategy that we extended from an existing work that used self-paced sampling strategy for residual point sampling for training a single PINN. We are not able to identify additional baselines for adaptive task sampling in meta-PINNs (without carrying out methodological developments as significant as the presented DATS strategy).

---

### Author Response · Authors · 2023-11-22
**Overall response (part 2)**

**Clarification about the validity of DATS-rp (reviewer Z9PQ):** We would like to provide clarifications to this important point raised by reviewer Z9PQ. DATS-rp actually follows the same assumption as derived throughout Eqs (7-10) because the training loss on each PDE configuration is calculated as the **sum** of PDE residual losses on all residual points for that PDE: therefore, p(\lambda) can be implemented as a weighting to the individual PDE training loss with equal number of residual points (p(lambda_i) \sum_{k=1}^{b_T/n} l_train), or as changing the number of residual points for individual PDE training losses (\sum_{k=1}^{p(lambda_i)*b_T/n} l_train).  To make this point more clear, we have revised Eqs (11-12) to include the **sum** of residual point losses in the PDE training loss.

**Clarification about overfitting and generalization (Reviewer USEY):** We would like to emphasize that generalization ability is a strong motivation of DATS – that’s exactly why the DAT’s optimization objective is to find task sampling probability to improve the PINNs’ performance on “validation” residual points. Furthermore, our experiments had a specific focus to assess the generalization capabilities of the method: in Section 5.3 of our paper, we delved into the performance of DATS when applied to new PDE configurations that were not included in the training dataset. Figure 6, in particular, displays the generalization capabilities of our approach by illustrating the L2 errors and Figure D7 in the appendix displays the disparity metrics for these unseen PDE configurations. The empirical evidence provided in these sections strongly suggests that DATS not only enhances the learning process but also ensures that the meta-model retains its predictive accuracy when faced with novel tasks.

**Clarification about improving sampling efficiency (Reviewer LK9J):** It is important for us to clarify that the motivation for DATS is not to improve the sampling “efficiency” in terms of computational cost of the meta-learning. Instead, the goal is to improve the sampling “efficacy” for meta-learning PINNs to improve the accuracy of the PINNs both for PDE configurations included in the meta-learning as well as for those outside the meta-training distributions (improving generalization). The presented results showed that, using the same budget for the number of residual points used, DATS improved the PINNs’ performance significantly with only a modest overhead to determine task difficulty ( a 3-8% increase in the overall training time). Furthermore, to achieve the same performance with uniform sampling, DATS can reduce the number of residual points to a range from less than 1% to 40% of what is needed using uniform sampling, ultimately reducing computational burden.

---

### Meta-Review · Area_Chair_usu6 · 2023-12-09

**Metareview:**

The reviewers reached an agreement for acceptance, and I am pleased to recommend this paper based on their expertise. It is essential that the authors address the reviewers' concerns in the final version.

**Justification For Why Not Higher Score:**

This proposed method just shows potential to more complex but only tested on some used PDEs.

**Justification For Why Not Lower Score:**

The meta learning for PINNs is well presented in both methodology and experimental results.

---

### Decision · Program_Chairs · 2024-01-16

Accept (poster)